# Organizational Geosocial Network: A Graph Machine Learning Approach Integrating Geographic and Public Policy Information for Studying the Development of Social Organizations in China

**Xinjie Zhao** [1,†] , **Shiyun Wang** [2,†] **and Hao Wang** [1,*]

1 College of Humanities and Development Studies, China Agricultural University, Beijing 100083, China;
wyh_cohd@cau.edu.cn

2 College of Economics and Management, China Agricultural University, Beijing 100083, China;
shiyunwang@cau.edu.cn

* Correspondence: haowang@cau.edu.cn

† These authors contributed equally to this work.

**Abstract:** This study aims to give an insight into the development trends and patterns of social organizations (SOs) in China from the perspective of network science integrating geography and public policy information embedded in the network structure. Firstly, we constructed a first-of-its-kind database which encompasses almost all social organizations established in China throughout the past decade. Secondly, we proposed four basic structures to represent the homogeneous and heterogeneous networks between social organizations and related social entities, such as government administrations and community members. Then, we pioneered the application of graph models to the field of organizations and embedded the Organizational Geosocial Network (OGN) into a low-dimensional representation of the social entities and relations while preserving their semantic meaning. Finally, we applied advanced graph deep learning methods, such as graph attention networks (GAT) and graph convolutional networks (GCN), to perform exploratory classification tasks by training models with county-level OGNs dataset and make predictions of which geographic region the county-level OGN belongs to. The experiment proves that different regions possess a variety of development patterns and economic structures where local social organizations are embedded, thus forming differential OGN structures, which can be sensed by graph machine learning algorithms and make relatively accurate predictions. To the best of our knowledge, this is the first application of graph deep learning to the construction and representation learning of geosocial network models of social organizations, which has certain reference significance for research in related fields.

**Keywords:** machine learning; geosocial network; graph structure; social organizations; big data; Chinese society

## 1. Introduction

With economic and social development, Chinese social organizations have been developing rapidly, participating in planning and governance, providing professional services in various fields such as health care, social security, and public education [1]. Although social organizations often work with or alongside government agencies, and may even receive funding or commissions from the government, they are actually independent third parties outside of the government in most domains.

When the People's Republic of China was founded, there were only about 100 national social organizations and 6000 local social organizations. Soon after the beginning of the Cultural Revolution in 1966 when the Ministry of the Interior, which was in charge of all Chinese social organizations, was abolished, social organizations almost vanished in mainland China. Thanks to the increasingly liberal social climate in China after the reform and

opening up, the announcement of the Regulations on Registration of Social Organizations and the Fund Management Measures laid a solid legal foundation for the development of social organizations, whose number nearly doubled in the following decade.

In the first decade of the 21st century, social organizations in China put on a spurt. Nowadays, however, confronted with a saturated market and continuously perfecting policies and legal systems, the growth rate has decreased (Figure 1), which indicates the shift of development philosophy in China, from the pursuit of speed to the pursuit of quality.

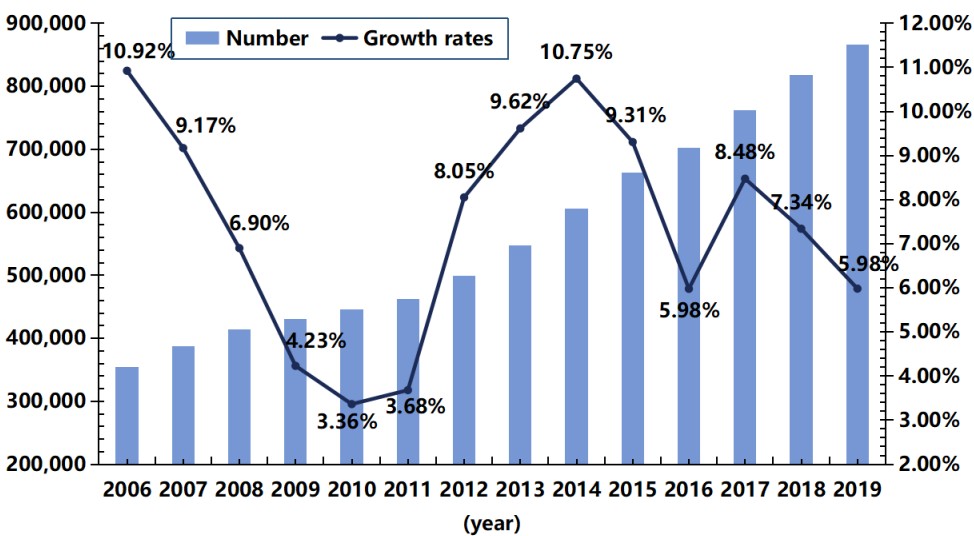

**Figure 1.** The development trend of social organizations in China before COVID-19 pandemic.

Social organizations in China can be divided into three categories: "top-down", "bottom-up", and "external imported". Government-run organizations and foundations are typical "top-down" social organizations. In contrast, the "bottom-up" social organization includes all kinds of local industry associations and private non-profit organizations. After China's accession to the World Trade Organization(WTO), the "externally imported" ones, whose funding, project operation and governance are mainly derived from foreign social organizations, is a force to be reckoned with, bringing new ideas and innovations to fields such as environmental protection, poverty alleviation and female rights. The vast territory, the uneven distribution of natural resources, the inter-mingling of various social classes, the unbalanced development and cultural diversity in China have contributed to the great differences in social development as well as the composition of social organizations from all-around China. Generally speaking, geographic location, including local economy, culture and policies, is an important factor in the growth of social organizations, and it's considerably crucial to explore the impact of abstract structures embedded in geographic information on the development of social organizations in China.

A social network is a structure composed of various social entities; the most familiar one to us is no doubt the Internet-based social network (e.g., Facebook, LinkedIn, or WeChat). However, except from individuals online, social organizations can also be an important composition of a social network [2]. This perspective provides a set of methods and theories for analyzing the structure of social entities as a whole, as well as explaining the patterns observed in these structures [3]. The social networks analysis(SNA) has recently become increasingly popular due to rising technology of graph machine leaning [4,5]. From the mathematical concept of graphs, the simple and straightforward function of graphs enables us to obtain a clearer picture of community structure and their interactions. However, previous literature paid little attention to the quantitative and structural exploration of organizational networks. In this paper, we accomplished the construction and exploratory analysis of specific machine learning algorithms and graph models by synthesizing political

and economic information embedded in organizational social network (OGN) based on real-world data.

Figure 2 illustrates the distribution of social organizations in China using the database constructed in this paper, revealing a nationwide organizational social network (OGN), where the dots represent social organizations of each administrative unit and the brightness of each dot represents its degree centrality. The concentration of social organizations is consistent with the distribution of prominent economic zones, such as the Yangtze River Delta and the Pearl River Delta. There is an imaginary diagonal line across China, called the Hu Line. The Hu Line has vast demographic significance and can also represent the distribution of social organizations: the number of social organizations on west side of the line is considerably lower than in those on the east.

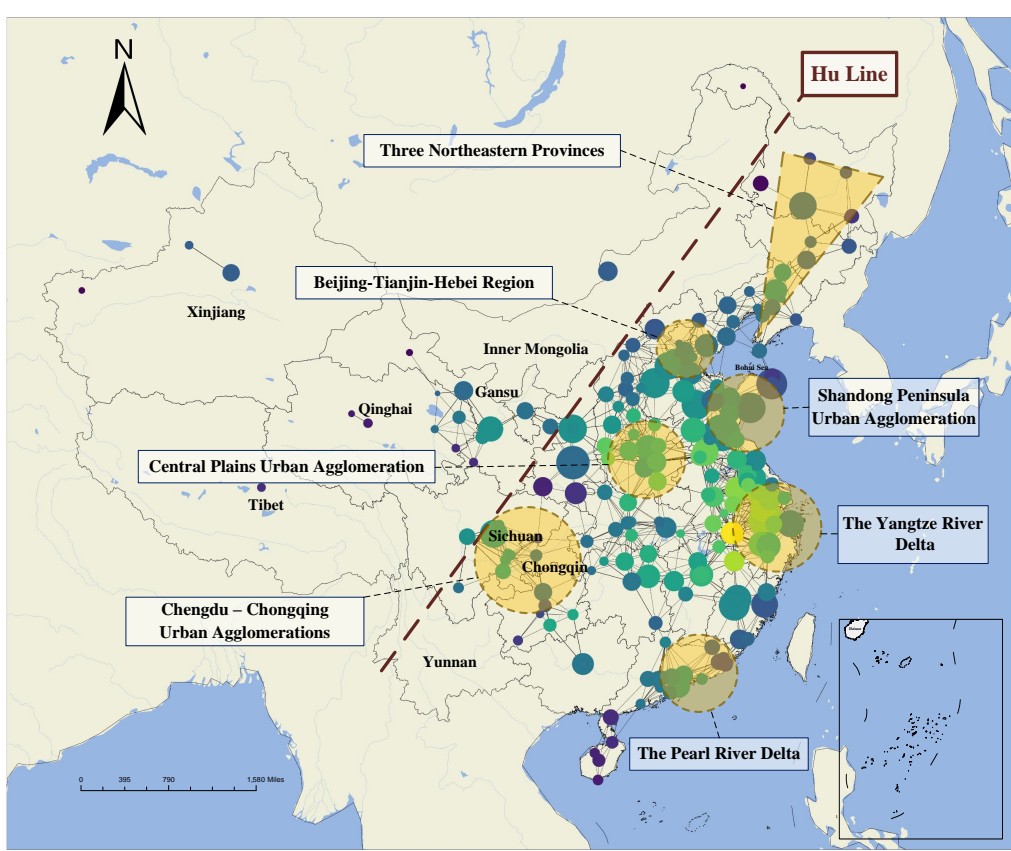

**Figure 2.** Distribution of social organizations in China.

The main contributions of this paper are as follows. Firstly, we used the open source data of the Ministry of Civil Affairs of China to construct a pioneering large-scale database of social organizations fusing public policy and geographic information, which is, to our knowledge, the first large-scale database of social organizations for research use. Secondly, we pioneered the application of graph structure to model the development of social organizations that integrate geographic information and public policy. Last, but not least, based on the graph attention mechanism, we propose a new graph attention network integrating textual information of social organizations, and apply it to the task of classifying graph networks based on geographic information and achieve a good result, laying a foundation for exploring the dynamic development model of regional social organizations.

The structure of this paper is organized as follows: Section 1 presents the introduction, with a brief history of social organizations in mainland China and main research ideas of the article. Section 2 introduces several research topics related to this research, including social networks, geographic information systems, natural language processing and graph neural network models. Section 3 focuses on the construction process of our brand new database

and some descriptive statistical analysis of the collected data. In Section 4, we propose four basic types of organizational social networks based on the theory of homogeneous and heterogeneous graphs, and attributed network embedding based on BERT and CNN. In Section 5, we investigate the organizational social network using graph machine learning models to explore the relationship between the network and geographic regions to which they belong. In Section 6, we draw conclusions for the paper.

## 2. Related Topics

### 2.1. Social Network

Since the 1990s, social networks have become an increasingly popular research topic, not only in social sciences, but also in computer science and physics. Social networks uncover the relations between social entities, as well as intrinsic social structures [6]. A traditional social network is an abstract structure that contains different relationships between people, such as friendship, common interests, and shared knowledge [7].

Location-based social network (Figure 3) is a variant of social networks that can create connections between abstract social networks and the real world environment by marking spatial information into the network. As in the Foursquare network, users can comment on events at the exact location where they occurred [8]. In the Twine network, for example, travel routes with GPS tracks are recorded and travel experiences are shared in a community [9].

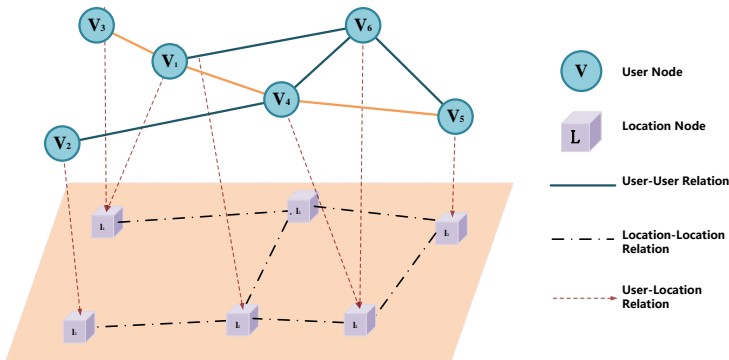

**Figure 3.** A concept map showing the structure of the location-based social network model .

Social network analysis considers individuals in a network, such as a person, a group, or an organization as nodes, with certain dependencies and collaborative relationships among them, which can be represented by connections between points, and the network is composed of nodes and their interrelationships [10]. This method takes the structural relationship between nodes as the guiding principle and considers that any action taken by an individual in the network comes from the individual's position in the social relationship structure system rather than an individual's motivation [11–13], i.e., the network position of the individual "forces" the actor to take a certain action [2]. Social network analysis can visualize the relationship between network members and the network structure and is often used to explore the key nodes in the network relationship [14].

### 2.2. Geographic Information System (GIS)

GIS is a computer-supported system which collects, stores, manages, retrieves, analyzes, and describes the location distribution of spatial objects and their related attribute data [15]. The word "geographic" in GIS does not refer to geography in a narrow sense, but refers to the spatial data, attribute data, and related data obtained on the basis of the geographic coordinate reference system in a broad sense.

Spatial data usually consists of three types of information: location, spatial relationships, and non-spatial attributes [16]. Location, namely, geometric coordinates, is used to determine the spatial position of spatial objects in the geographic coordinate system. Spatial

relations describe the spatial connections between spatial objects, mainly covering metric relations, such as distance between spatial objects, extension relations, or orientation relations, which indicate the orientation between spatial objects. Topological relations indicate the relationship between spatial objects, such as connectivity or adjacency. Non-spatial properties are properties that are not relevant to geometric position. The establishment and data mining of a spatial database is an important research direction in GIS, and Figure 4 shows us the idea of geographic information mining for social organizations in this paper [17].

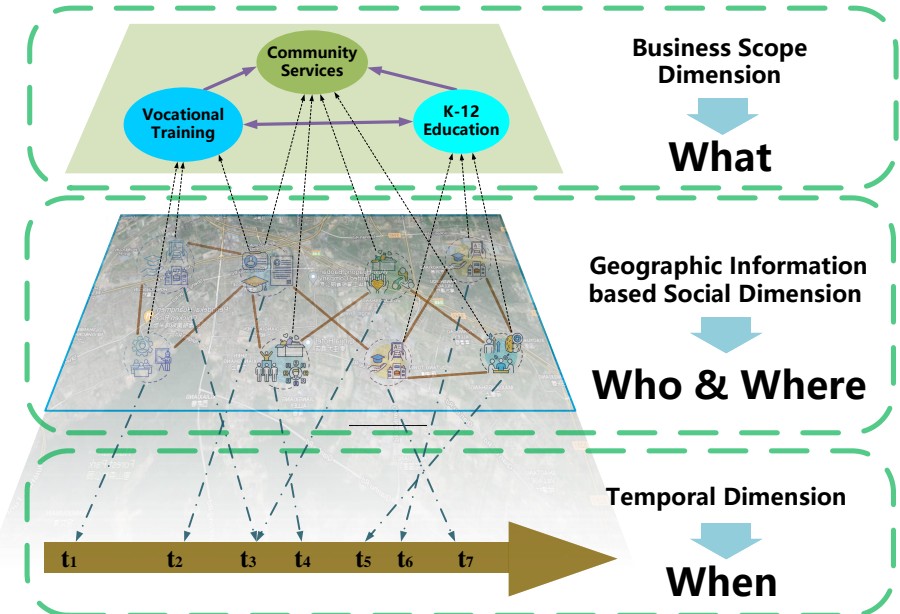

**Figure 4.** A concept map that has analyzed the geographic information system of social organizations.

*2.3. Natural Language Processing*

The language people use to communicate in daily life is natural language, and so is the text content in the dataset we construct. Text is relatively standardized, with relatively complete grammatical and syntactic and structural information. The goal of Natural language processing (NLP) is to bridge the gap between natural language and machine language [18], using calculation power to analyze the structure and syntax of natural language and extract information from the text content [19]. The main categories involved in natural language processing are word division, lexical annotation, syntactic analysis, sentiment recognition, automatic translation, text summarization [20], knowledge graph [21], and so on.

English text has a natural advantage because each word is separated from each other by a space, while for Chinese text, there is no division among words; furthermore, Chinese text needs to be divided to form a separate word order [22]. The emergence of word-splitting tools has lowered the threshold for high-quality word splitting of text; Jieba is a easy-to-use word splitting tool for Chinese text [23].

The lexical properties refer to the basic attributes of words, and lexical annotation is the process of marking words with names, gerunds, adjectives, adverbs, or other lexical properties. The lexical annotation with machine learning is mainly performed by using some feature values extracted from the data by neural networks. In recent years, deep learning models such as convolutional neural networks and LSTM (long short-term memory network) have also been used for lexical annotation. We choose the BERT model, which is built on top of the transformer and has powerful language representation and feature extraction capabilities. For a given text corpus, the input representation consists of a word vector, a segmented embedding vector, and a positional embedding vector summation, which is then passed through a bidirectional transformer encoder to obtain the corresponding text word vector output. Its extended models are mostly based on its model architecture

to design novel language learning tasks, and then trained on domain-specific large-scale text corpus to obtain new models.

*2.4. Graph Machine Learning*

Since the recent research focus on graph-structured data, a variety of machine learning algorithms have been proposed for representation learning in graphs, which, based on whether the labeled data are available, can be generally divided into three main categories [24]: network embedding (such as graph autoencoders), graph regularized neural networks, and graph neural networks (Figure 5).

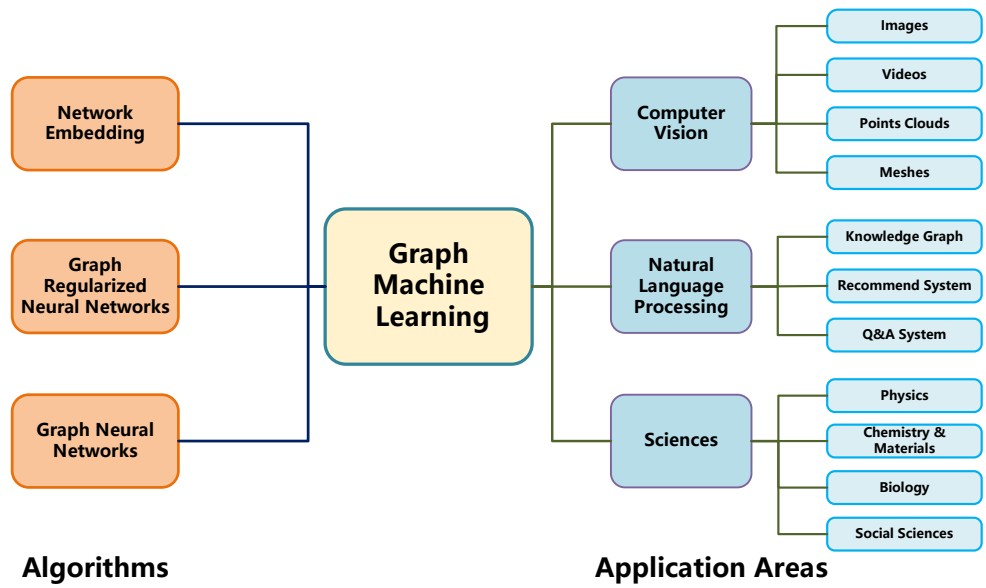

**Figure 5.** An illustration of graph machine learning.

Starting directly from the structure of graphs, a graph neural network (GNN) [25] proposes aggregated and combined models aiming to learn differentiable functions over discrete topologies with arbitrary structure [26].

Most of the early graph neural network models [27] use recurrent neural structures to propagate information about neighbors and select generations until they reach a stable immobile point to learn the representation of the target node. The classical formulation of graph neural networks is as follows:

$$h_u^t = \sum_{v \in N(u)} f\left(x_u, x_{(u,v)}^e, x_v, h_v^{t-1}\right) \tag{1}$$

where $h_u^t$ denotes the state of node $u$ at the $t$th recursion; $f(X)$ denotes the recursive function; $N(u)$ denotes the set of neighboring nodes of node $u$ in the graph; $x$ denotes the feature. The initial state of $h_u^0$ is a random value, and $h_u^t$ consists of the features $x_u$ of the node $u$ itself and the edge features of the neighboring nodes $v$. $x_{(u,v)}^e$ is the feature $x_v$ of the neighboring node $v$, and $h_v^{t-1}$, at $t-1$ times of generation selection. This has the advantage that the formula can be generalized to all nodes in the graph, without the constraints of inconsistency in the number and order of neighboring nodes, and it also gives the graph neural network the ability to process recurrent graphs. However, these studies are computationally expensive, and immobility hinders the diversity of node distributions, which is not conducive to fully learning the representation of nodes.

### 2.4.1. Graph Convolutional Neural Network

Later, based on the spectral analysis of researchers who defined the convolution operation on the graph [28], the graph convolution network (GCN, graph convolution network) came into being.

A graph convolutional neural network (GCN) is a fusion algorithm that applies graph structure data to traditional convolutional neural networks (Figure 6), and as a powerful tool for extracting features, it can make good use of neighborhood graphs constructed in a simple KNN so that the learned feature representation contains two different types of information: feature information of the sample nodes and their associated neighborhoods.

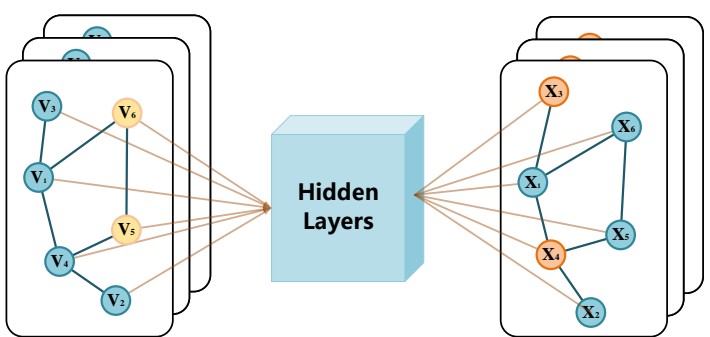

**Figure 6.** Basis structure of GCN.

A common graph deep neural network consists of a cascade of multiple graph convolution layers, each of which can be represented as

$$Z_{\text{gcn}}^{k} = \text{RELU}\left( \tilde{A}_{\text{sym}} Z_{\text{gcn}}^{k-1} W^{k} \right) \tag{2}$$

$Z_{\text{gcn}}^{k-1}$ denotes the feature of the $k-1$th layer, $Z_{\text{gcn}}^{k}$ denotes the feature of the $k$th layer. $\tilde{A}_{\text{sym}}$ is the normalized adjacency graph matrix, $W^{k}$ denotes the parameters of the $k$th layer of the graph neural network, and *RELU* denotes the activation function. Assuming that the activation function is not considered and the weight matrix is ignored, we can obtain $\lim_{k\to\infty}\left(\widetilde{A_{\text{sym}}}\right)^{k} H^{0} = H^{\infty}$. This means that $H$ depends only on the degree of the nodes, which indicates that as the number of layers increases, the model loses the discriminative information provided by the node features, and therefore the features appear to be oversmoothed. Therefore, when the number of layers of the network deepens, the final features learned by the graph neural network lose the uniqueness of the sample points themselves, which affects the performance of clustering.

### 2.4.2. Graph Attention Neural Network

A graph attention network (GAT) is a graph neural network architecture proposed by Petar Veličković et al. [29], which improves the classical graph neural network by combining graph convolution and attention mechanism.The basic structure of GAT is shown in Figure 7. GAT computes the attention score on the input graph, which represents the importance of the input mapping to the output state. Self-attention is introduced to determine the attention score of the input graph preprocessed by GCN . When each node updates the output of the hidden layer, attention is computed on its neighboring nodes. Each node and its neighboring nodes compute attention in parallel and can assign arbitrary weights to neighboring nodes.

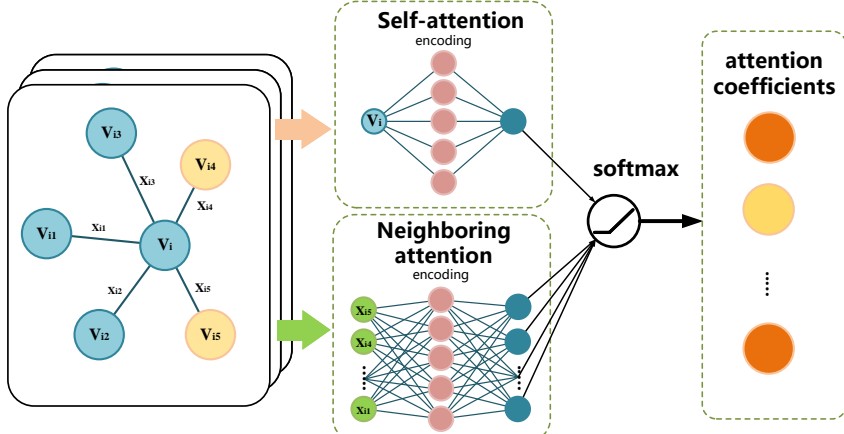

**Figure 7.** Structure of graph attention neural network.

Graph attention networks have a wide range of applications in social sciences; Weiping Song et al. [30] modeled social interactions among pedestrians by graph attention networks to predict their trajectories. V. Kosaraju et al. [31] constructed dynamic graph attention neural networks to build online community recommendation systems based on dynamic user behavior and environment-related social influences. J. Piao et al. [32] predicted socioeconomic relationships among customers by considering their demographics, past behaviors, and social network structure.

In view of the previous research on graph attention networks in social sciences, this paper uses graph attention networks as a social organization network structure feature extraction layer to learn social organization network graph features.

## 3. The Novel Database of Social Organizations in China

In China, public access to information related to social organizations can be browsed online through the National Social Organization Credit Information Public Platform (hereafter, the Platform; https://xxgs.chinanpo.mca.gov.cn/gsxt/newList, accessed on 17 May 2022), supervised by the Ministry of Civil Affairs. The Platform stores all of the basic information entries of each organization, Figure 8 is an example.

However, users can only search for information about one specific organization by entering keywords or the exact social credit code, and can only search for one organization at a time, which severely limits the amount of data that researchers can access for research purposes. Furthermore, users have to pass a human–machine verification operation before every single search. In China, where tens of thousands of social organizations are established every year and the Platform stores all of their basic information, if we try to manually perform the acquisition of all social organizations, millions of searches and downloads are required, which is a huge drain in terms of manpower, money, and time, thus limiting or even preventing the role of big data analysis of social organizations in China. Therefore, the use of web data scraping methods for bulk collection and collation of web data is a must.

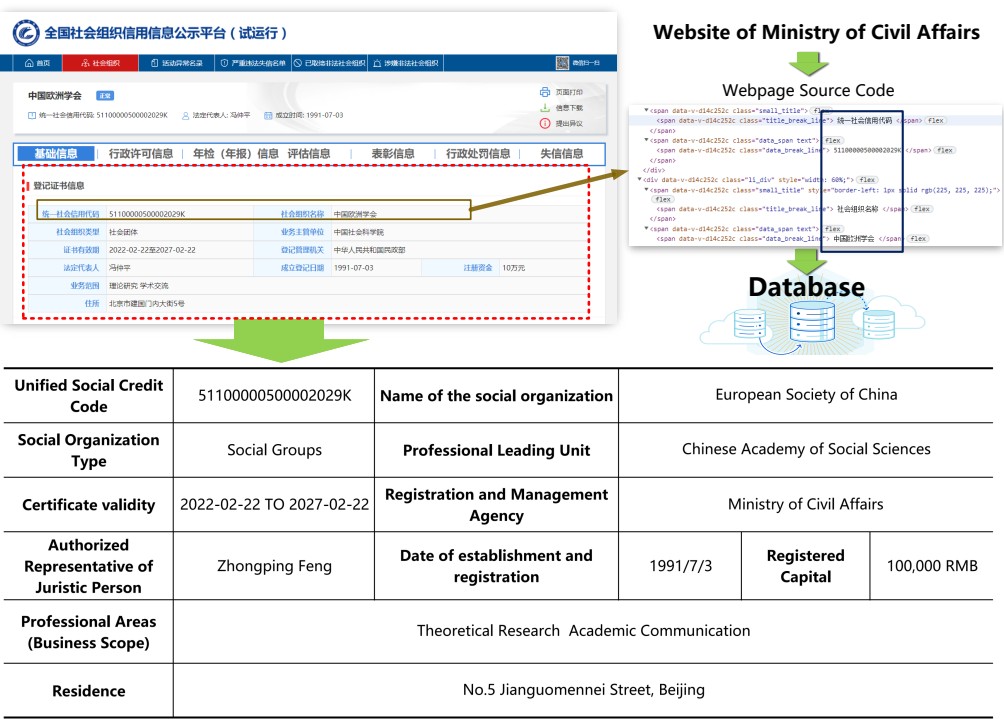

| Unified Social Credit Code | 51100000500002029K | Name of the social organization | European Society of China | | |
|---|---|---|---|---|---|
| Social Organization Type | Social Groups | Professional Leading Unit | Chinese Academy of Social Sciences | | |
| Certificate validity | 2022-02-22 TO 2027-02-22 | Registration and Management Agency | Ministry of Civil Affairs | | |
| Authorized Representative of Juristic Person | Zhongping Feng | Date of establishment and registration | 1991/7/3 | Registered Capital | 100,000 RMB |
| Professional Areas (Business Scope) | Theoretical Research  Academic Communication | | | | |
| Residence | No.5 Jianguomennei Street, Beijing | | | | |

**Figure 8.** A flow chart showing the database construction based on open data platform of the Ministry of Civil Affairs of China.

### 3.1. Design and Implementation of Web Crawlers

In this paper, we have written a web crawler with data processing program using Python. The web crawler accesses web pages through hypertext transfer protocol (HTTP). The web crawler generally sets the starting set of seed URLs at the beginning, and after establishing a successful connection with the seed URL server, it parses the contents of the corresponding web pages to obtain all the URLs that can be linked from them [33]. It then searches the web page and downloads the target data, which, as is shown in Figure 8, may be encoded in Hypertext Markup Language (HTML) or obtained through links to JS codes. The number of pages visited and searched depends on the parameters set in the program prior to startup. New URLs are then added to the queue to be crawled until the termination conditions are met, and then the parsed results are stored. The crawler we designed fully complies with the prescribed robots protocol and sets the request information for legal requests. The final step is to transform the data and integrate it into a structure suitable for analysis, and the obtained data in Datafram format are saved as CSV files to the cloud for subsequent calls.

As seen in Table 1, each web page contains the details of a specific social organization. After using regular expressions to obtain the body information, we can obtain the text information easily. However, difficulties in the design and writing of the web crawler program lie in how to crack the encryption of the web URLs (Figure 9), skipping the human–machine verification and searching process, and directly obtaining the web address of each social organization point-to-point.

Through the collection and collation of the basic components of social organizations, which are shown in Table 1, data cleaning was carried out to establish a database of social organization. As of January 2022, we have accessed a total of 1.09 million social organizations and their related information. We declare that the data obtained in this study are public and for research use only, without any commercial and malicious behavior. In addition, for legal reasons, we do not publish the exact technical details of how to break the encryption on the website.

**Table 1.** The detailed elements published in the platform can be used as the basic variables that constitute the database.

| Name | Type | Content |
|---|---|---|
| Name | Text | The name of the social organization that the legal person has registered. |
| Unified Social Credit Code | String | Unified Social Credit Code is an 18-digit number (sometimes including letters), unique for every single Chinese organizational. |
| Social Organization Type | Text | The type of the social organization that the legal person has registered. |
| Certificate Validity | Date | The validity period is the time interval during which the registration certificate is guaranteed to maintain its status information. |
| Date of Establishment and Registration | Date | The date of establishment and registration. |
| Authorized Representative of Juristic Person | Text | The authorized representative of juristic person is one who represents or stands in the place of another under authority recognized by law especially with respect to the other's property or interests. |
| Professional Areas (Business Scope) | Text | Related trade, scientific or other professional areas of this social organization including charitable activities, poverty alleviation, environment protection, education, etc. |
| Residence (Location and Coordinates) | String | The residence is the designated place to carry out social organization-related activities. |
| Professional Leading Unit | Text | State Council relevant departments and local government relevant departments at county level and above, or organs empowered by the State Council or local government at county level and above, serve as the relevant leading units of social organizations in related trade, scientific or other professional areas. |
| Registration and Management Agency | Text | The basic peoples government agencies for registration and management of social organizations, namely the Ministry of Civil Affairs and local Civil Affairs departments at county level and above . |
| Registered Capital | Integer | The registered capital is the capital contributed or promised to be contributed by all shareholders when they apply to the local industrial and commercial bureau for the establishment of a social organization. |

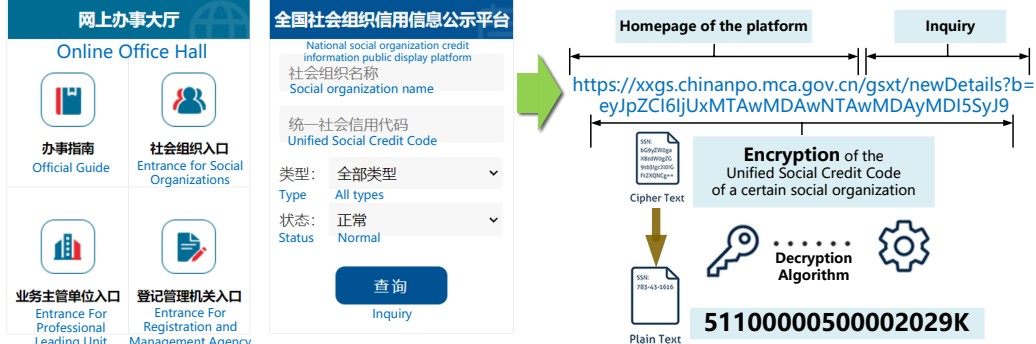

**Figure 9.** Composition rules and decryption of target URLs.

*3.2. Data Cleaning and Geographic Information Integration*

The quality of data plays a key role in the results of data mining. Data cleaning usually includes dealing with missing values and redundant values, as well as noise. The text collected by web crawlers is mostly unstructured data containing data noise. By observation, we found that there was a certain percentage of noise in the acquired data, which is of no help for understanding the semantics of the text. We deduce that, since the Platform of the Ministry of Civil Affairs only serves as a tool for integrating and publishing information, and detailed data are filled in and uploaded by local civil affairs departments, problems and errors may arise during the uploading process, such as meaningless symbols or tags, JS codes, traditional or abandoned Chinese characters, line breaks, different time formats, and so on, so we need to clean and standardize the obtained data and integrate the relevant geographical information of each social organization to provide a high-quality build of a complete and usable database for research use.

After normalizing the temporal data, the study of the temporal dimension could be carried out. For example, Figure 10 uses the data of the registration time of the organizations. Among the established social organizations, 50,774 have been in existence for less than one year, 152,661 have been operating for one to three years, 155,881 have been operating between three and five years, the largest proportion of social organizations have been functioning for five to ten years, and even more than 240,000 have been running for more than 10 years.

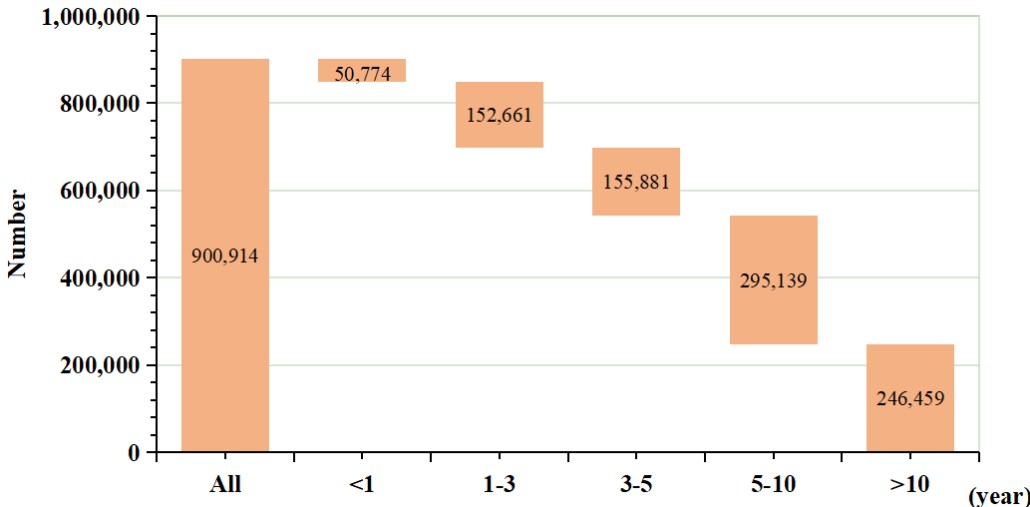

**Figure 10.** Social organizations categorized by the time of operation.

Meanwhile, the geographical information of social organizations can be obtained by two different methods. The first one is to use the registered address information contained in the database, by calling the API to search and obtain its precise latitude and longitude coordinates which, however, is relatively time-consuming and cannot be applied on a large scale. There is another method which we reckon is a more efficient way to categorize the locations directly according to the coding rules of the unified social credit code. As is shown in Table 2, the unified social credit code, a unique, 18-digit national registration number, follows a standard pattern, which means that we can directly use the 6-digit area code embedded in the unified social credit code to locate social organizations down to the exact administrative division of the county where they are located.

**Table 2.** The composition rules of the Chinese social unified credit code.

| The Unified Social Credit Code | | | | |
|---|---|---|---|---|
| 1st | 2nd | 3rd–8th | 9th–17th | 18th |
| 5 | X | $X_1\ X_2\ X_3\ X_4\ X_5\ X_6$ | X X X X X X X X X | X |
| Registering Authority | Registered Entitiy Type | Registered Region | Organization Code | Check Digit |
| 1: Public institutions<br><br>5: Social organizations<br><br>9: Enterprises<br><br>Y: Others<br><br>The first digit of social organizations starts with 5 . | 1: Social groups<br><br>2: Non-enterprise Private Units<br><br>3: Foundation<br><br>4: Others<br><br>The second digit of the code indicates the type of the social organization. | This string of six numbers indicates where the organization is registered. | The same as the China Organization Code certificate | This singular entry is a safety measure that allows Chinese authorities to confirm that the code is indeed valid. |

After obtaining the basic geographical information of social organizations, we can explore and study social organizations in the spatial dimension. The map in Figure 11 displayed here shows how the number of newly established social organization varies by province. The shade of the province corresponds to the magnitude of the indicator.The darker the shade, the higher the value.

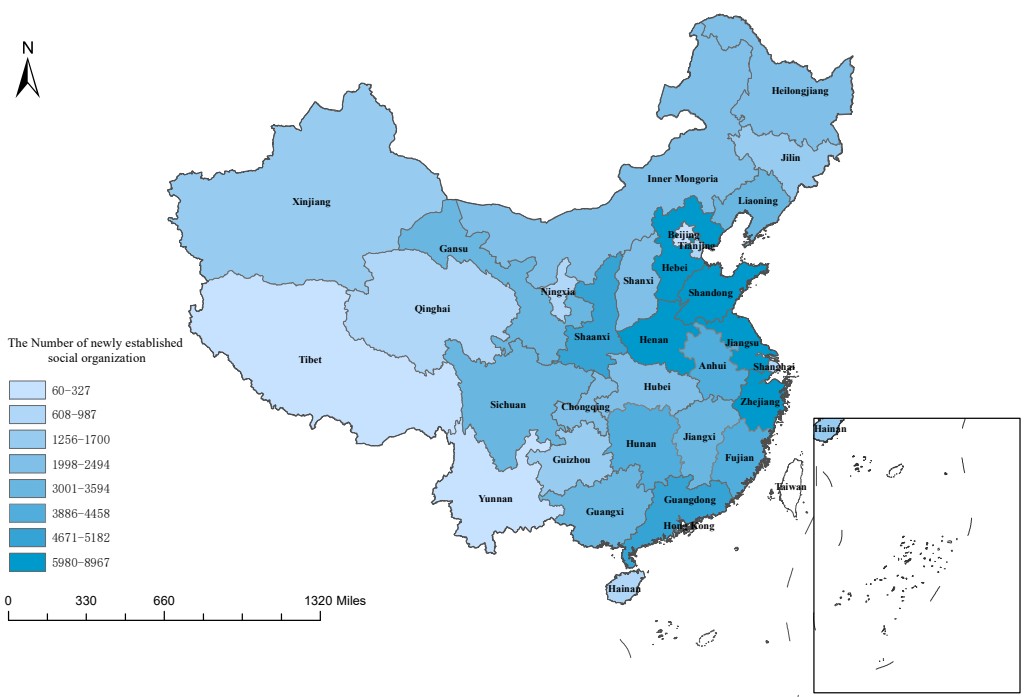

**Figure 11.** The number of newly established social organizations between August 2020 and August 2021.

### 3.3. Text Data Analysis

Since most of the information in the database is Chinese text, how to obtain and analyze the features and semantic information of the Chinese text is of great significance to our study, which would determine the research direction. We firstly performed a basic word

separation process on the names of social organizations and their business introduction in the database.

Table 3 shows us clearly the frequency of the occurrence of high-frequency words of different lexicons, enabling us to have a more intuitive sense of the development of social organizations in China. The first line of each cell is the Chinese translation of the word, the second line in parentheses is the original Chinese text, and the third line in italics is the number of times the word appears. The shade of the cell corresponds to the magnitude of the indicator. The darker the shade, the higher the value. In the listed categories, *vn* refers to the gerunds, *n* refers to the noun, *s* refers to the preposition, *nl* refers to the noun idiom, and *adj* refers to the adjective.

**Table 3.** Top 10 popular keywords ranked by the occurrence frequency.

| *n* | *vn* | *nl* | *adj* | *s* |
|---|---|---|---|---|
| Association (协会) *266,285* | Training （培训） *40,737* | Kindergarten （幼儿园） *158,868* | Small （小） *8882* | East （东方） *3750* |
| Schools (学校) *77,611* | Education （教育） *28,703* | Pension （养老） *21,150* | New （新） *8561* | Overseas （海外） *853* |
| Service (服务中心) *67,972* | Poverty Alleviate （扶贫） *27,758* | teenager （青年） *10,856* | Big （大） *3758* | Outside the Party （党外） *809* |
| Town (镇) *58,705* | Mutual support （互助） *24,239* | Sunshine (阳光) *9880* | Old （老） *3215* | City North （城北） *771* |
| Community (社区) *49,468* | Organize （组织） *21,857* | Charity （慈善） *6048* | Healthy （健康） *3036* | Off-campus （校外） *595* |
| Street (街道) *46,462* | Development （发展） *18,581* | Nursing homes （养老院） *3966* | Happy （幸福） *2351* | External （对外） *529* |
| Centre (中心) *39,818* | Promote （促进） *14,079* | High-tech Zone （高新区） *3617* | Peace （平安） *1774* | Outdoors （户外） *523* |
| Village (村) *35,122* | Planting （种植） *4643* | Blue Sky （蓝天） *3158* | Hard （难） *1057* | Haikou （海口） *471* |
| District (区) *28,118* | Exercise （健身） *4046* | Baby （宝贝） *2979* | Harmonious （和谐） *942* | West Coast （西海岸） *427* |
| Culture (文化) *25,834* | Care （关爱） *3050* | Gold （金） *2683* | Good （好） *589* | Suburbs （郊区） *354* |

Table 3 reveals that the nouns in the results are all suffixes of certain words. The words "kindergarten" and "school" appearing after "association" is a reflection of the current boom in China's education market. It corresponds to the fact that private education in China as the essential form of social forces has developed rapidly and accumulated effective experience in the dissemination of knowledge. Note that the gerunds "poverty alleviate" is in first place, which infer that the Chinese government focuses on improving the living conditions of poor households and helping poor areas to develop production and change the face of poverty, while social organizations, as a third-party force, complement the synergistic effect of multi-subject governance. Similarly, we notice that the word "pension" is in second

place and "nursing homes" is in sixth place, reflecting the serious aging situation in China and the active participation of social organizations in the pension business.

## 4. Graph Model in Organizational Social Networks

### 4.1. Overview of the Graph Structure

Data exist in a plethora of different forms and sizes, but most of them can be presented as two types: structured data and unstructured data (Figure 12).

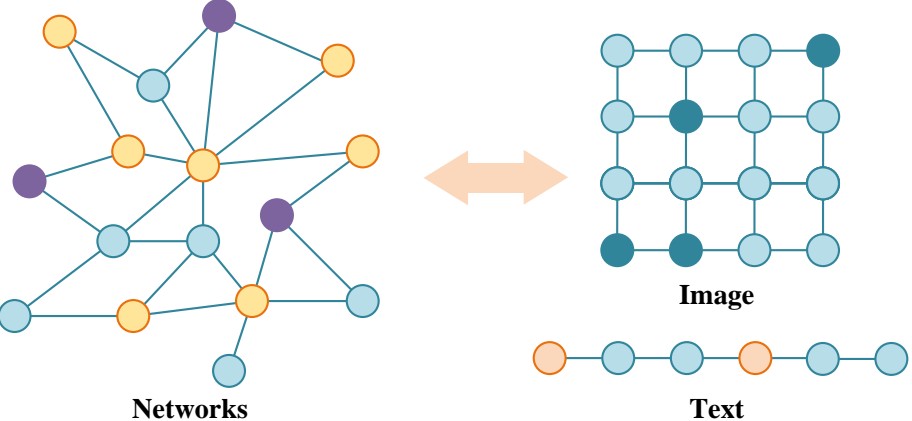

**Networks**　　　　　　　　**Text**

**Figure 12.** Euclidean structure data and Non-Euclidean structure data.

Structured data, for example, temperature, names, dates, stock information, location, and pictures, comprise clearly defined data types with patterns in a standardized format that enable them to organize searchable information efficiently. Modern machine learning algorithms have achieved amazing performance in processing structured data (such as AlphaGo [34], ResNet [35], etc.).

Graph, a typical unstructured data, is more flexible and variable compared with structured data, which, at the same time, makes it relatively more difficult to perform machine learning tasks on graph structured data. However, due to the wide application of graph models in human society, it is of great importance to study graph and related machine learning algorithms. One of the most vivid applications of graph structured data is the virus transmission models being used to characterize the transmission pattern of viruses across countries constructed during the COVID-19 pandemic [36] , which played a huge role in controlling the spread of epidemics.

A graph $G = (V, E)$, consisting of two sets, nodes $V$ (also called vertices) and edges $E$ (also called arcs), is able to represent entities and their relations in the graph structured data. An edge $e_{ij} = (u_i, u_j) \in E$ represents an edge pointing from $u_j$ to $u_i$, and the neighboring nodes of node $v$ are defined as $N(v) = \{u \in V \mid (v, \quad u) \in E\}$. The adjacency matrix A is a matrix of size $n \times n$; $n$ represents the number of nodes in the graph. If there exists an edge connecting nodes $u_i$ and $u_j$, then $A_{ij} = 1$, otherwise $A_{ij} = 0$. A node in a graph has attributes or features $X \in R^{n \times d}$ which is the attribute matrix of the node, or called the feature matrix of the node, where $X_v \in R^d$ represents the attribute vector of the node $v$. A graph may also have attributes of edges $x^e$ , $X^e \in R^{m \times c}$ is the attribute matrix of edges, where $X_{v,u} \in R^c$ represents the attribute vector of edge $(v, u)$, and $c$ represents the dimension of the attribute. The attributes and features represent the same meaning.

### 4.2. Homogeneous Networks of Organizations

Homogeneous networks, which use a single network architecture, have the same node and link types. Homogeneous networks are network structures composed of the same kind of nodes and link types.

As shown in Table 4, we introduce two types of homogeneous networks: competition and cooperation networks, and supply-chain networks. Each of these types is potentially useful in modeling social organizations and their relationships.

**Table 4.** Homogeneous networks.

| Network Type | Features | Graph Structure |
|---|---|---|
| **Competition and Cooperation Networks** | A competition network [37]consists of market (direct and in-direct) competitors and all the active, competitive ties among them. The Resource Based View theory [38] assumes that competitive or cooperative relationships between organizations is the result of resource gaps To gain access to resources and competencies available inside the network held by other actors [39] the organizations ought to be able to develop and utilize inter organizational relationships The trade-off between cooperation and competition has been emphasized as a mean of creating progress among actors involved in long-term relationships. A Competition and Cooperation network can be represented as an undirected graph $G_C = (V, E_C, s)$ , where $V$ denotes the set of social organizations that are related with each other, and the edges $E$ denote the link type, such as competition, cooperation or even coopetition, between two organizations. and $c(v_1, v_2)$ indicates the conflicting degree of credibility values of node $v_1, v_2$. |  |
| **Supply-Chain Networks** | Businesses are part of a complex network formed within organizations, through supply-chain network, we can highlight the interactions between organizations, such as the cross-organizational flow of employees, information and raw materials [40]. Instead of remaining static, the supply-chain network is dynamic, adaptive and constantly evolving, by modeling it, we can predict and guide an organization's business strategy more effectively and minimize risk at a system level. An supply-chain network in the business layer can be represented as a directed graph $G_S = (U, E_S, p, t)$, where $U$ and $E$ are the node and edge sets, respectively. Nodes $u \in U$ represent different types of social organizations, such as vocational education institutions or nursing homes , which can provide/receive services or propagate information at time $t_i \in t$. A directed edge, $(u_1 \rightarrow u_2) \in E_S$, between nodes $u_1, u_2 \in U$ represents the direction of service delivery or information propagation. Each directed edge $(u_1 \rightarrow u_2)$ is assumed to be associated with an service delivery or information propagation probability, $p(u_1 \rightarrow u_2) \in [0, 1]$. |  |

*4.3. Heterogeneous Networks of Organizations*

Heterogeneous networks have a different set of node and link types. The advantages of heterogeneous networks are the abilities to represent and encode information and relationships from different perspectives. During the development process of social organizations, different types of social entities are involved, for example, government, policymakers, policies, services, community members, and, of course, social organizations. Table 5 below provides two types of heterogeneous networks for modeling the relationships between social organizations and other social entities: policy networks and service networks.

**Table 5.** Heterogeneous networks.

| Network Type | Features | Graph Structure |
|---|---|---|
| **Policy Networks** | Public organizations are engaged in interdependent relations with the public administration [41]. The policy networks can represent the correlations among different types of social entities, such as governments, policymakers, policies, and social organizations, during process of social development [42]. The multisectoral and intersectoral characteristics of the interactions between these social entities have potential to promote the evolution of social structure and well-being for all. A policy network $G_I = (\{P, U, V\}, E_I)$ consists of nodes representing governments, organizations, policymakers, and policies, and the edges $E_I$ indicate the interactions among them. For example, edge $(pßv)$ demonstrates that policymakers $p$ issue a policy item $u$, and $(vßu)$ represents that this policy $u$ is accepted by organizations $u$. | |
| **Service Networks** | Centered on the provision of public services for community members [43], a service network can be represented as a directed network $G_S = (\{U, S, V\}, E_S)$, where the nodes can be all kinds of organizations and community members, and the edges $E_S$ denote the link type between two nodes, namely, the *services* provided by social organizations to local residents, and people's attitudes towards these services, such as supporting, denying, opposing, etc. Service networks can be used to measure the contribution and impact of social organizations to the society. | |

### 4.4. Attributed Network Embedding with Text Information

In addition to the structural features of the social organization network, the text content in the database, such as name, business scope, registered capital, and so on, needs to be processed in order to obtain the basic information of the social organization before being input into the machine learning model (Figure 13).

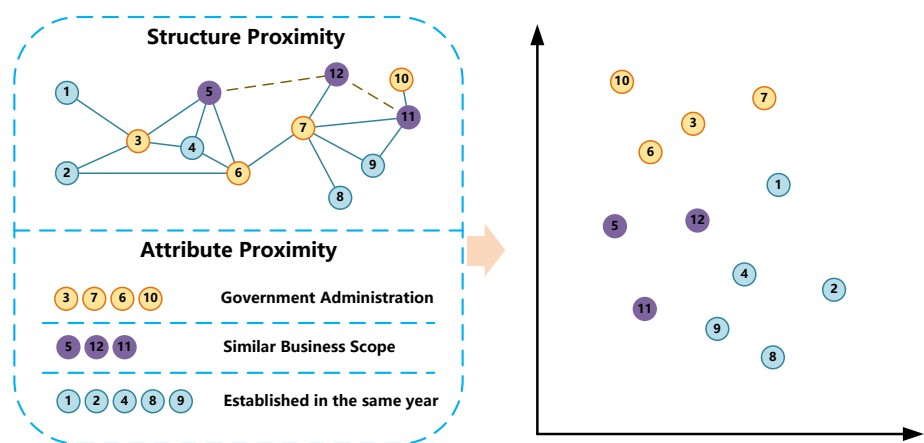

**Figure 13.** Attributed social network embedding.

In this paper, the length of the text content is limited to $L$. If the length of the text content exceeds $L$, then the excess part would be truncated, while if the length of the text content is less than $L$, placeholders would be used to fill the text until the length is $L$. $x_j^i \in \mathbb{R}^d$ denotes the word vector of the $j$th word in the text $p_i$, so the vector of the text $p_i$ can be expressed as $X_{1:L}^i = [x_1^i; x_2^i; \ldots; x_L^i]$ where $X_{1:L}^i \in \mathbb{R}^{L \times d}$, $x_1^i$ denotes the word vector

of the second word in the text $p_i$, $x_2^i$ denotes the word vector of the second word in the text $p_i$, and $x_L^i$ denotes the word vector of the *L*th word in the text $p_i$ (Figure 14).

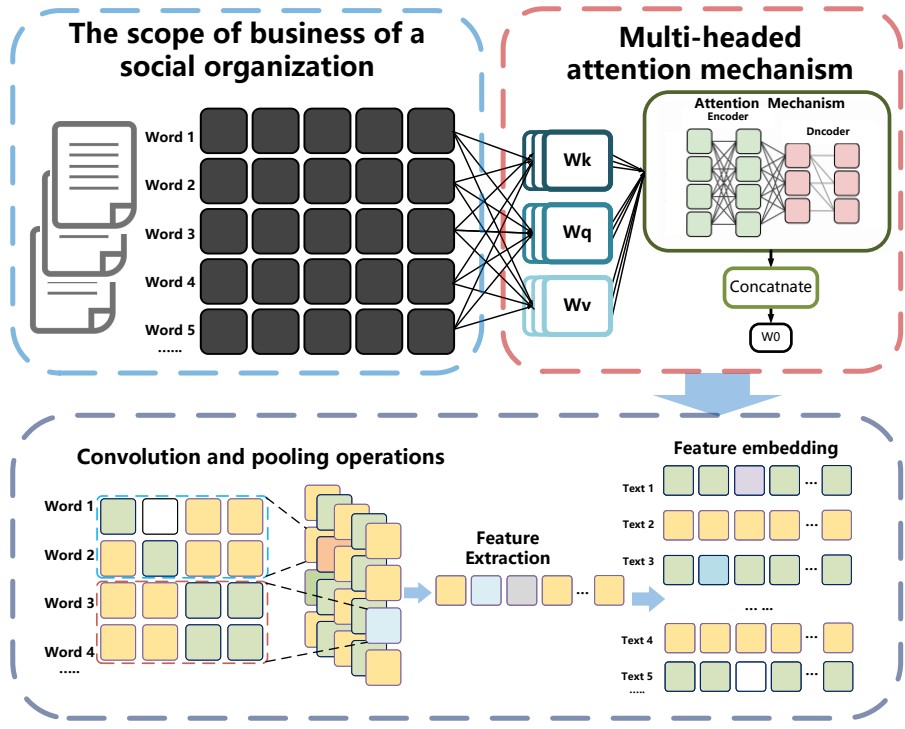

**Figure 14.** Attention mechanism: natural language processing.

4.4.1. Multi-Headed Self-Attention Mechanism

In the next step, we adopt a multi-headed self-attentive mechanism to update the word vectors in the text content of each social organization in the database. The multi-headed self-attentive mechanism can explore the connections among word vectors from different perspectives, thus improving the expressiveness of word vectors. $h$ denotes the number of heads of the self-attentive mechanism. Consider a self-attentive mechanism with $h$ heads; $j$ denotes the ordinal number of the head, and the three input matrices of the self-attentive mechanism for the $j$th head are denoted as query matrix $Q_j \in \mathbb{R}^{L \times \frac{d}{h}}$, matrix $K_j \in \mathbb{R}^{L \times \frac{d}{h}}$, and the value matrix $V_j \in \mathbb{R}^{L \times \frac{d}{h}}$. Taking the embedded vector of text $p_i$,

$$X_{1:L}^i = \left[ x_1^i; x_2^i; \ldots; x_L^i \right], \tag{3}$$

as an example: For simplicity, we use $X$ to denote $X_{1:L}^i$, then we have $K_j = XW_j^K$, $Q_j = XW_j^Q$ and $V_j = XW_j^V$, where $\left\{ W_j^Q, W_j^K, W_j^V \right\} \in \mathbb{R}^{d \times \frac{d}{h}}$, $W_j^K$ denotes the parameter matrix corresponding to the key matrix of the $j$th head in the self-attentive mechanism, $W_j^Q$ denotes the parameter matrix corresponding to the query matrix of the $j$th head in the self-attentive mechanism, and $W_j^V$ denotes the parameter matrix corresponding to the value matrix of the $j$th head in the attention mechanism. The output of the $j$th head of the self-attentive mechanism is represented as

$$Z_j = \text{Attention}\left(Q_j, K_j, V_j\right) = operatornamesoftmax\left(\frac{Q_j K_j^T}{\sqrt{d}}\right) V_j \tag{4}$$

where $Z_j \in \mathbb{R}^{L \times \frac{d}{h}}$. In this paper, the output of the $h-$headed self-attentive mechanism is expressed as $Z = [Z_1; Z_2; \ldots; Z_h]$, $Z_1$ is the output of the self-attentive mechanism for the 1st head, $Z_2$ is the output of the self-attentive mechanism of the 2nd head, and $Z_h$ is the output of the self-attentive mechanism of the $h$th head, then we have

$$Z = \text{MultiHead}(X, X, X) = \text{Concat}(Z_1, \ldots, Z_h)W^0 \tag{5}$$

where $Z \in \mathbb{R}^{L \times d}$, $W^0 \in \mathbb{R}^{d \times d}$, and $W^0$ denotes the parameter matrix of the $h-$head self-attentive mechanism.

4.4.2. Convolutional Neural Networks and Pooling Operations

Then, we use CNN and pooling operations to obtain semantic information from the text contents in the database. We use convolution kernels to perform the convolution operation $W \in \mathbb{R}^{k \times d}$ on the text vector $X^i_{e:e+k-1}$, where $X^i_{e:e+k-1}$ denotes the $e$ th word vector to the $e + k - 1$ th word vector in the text content $p_i$; and $k$ denotes the perceptual field size of the kernel. For all word vectors in $X^i_{e:e+k-1}$, the convolution operation can be expressed as

$$t_j = \sigma\left(W * X^i_{e:e+k-1} + b\right) \tag{6}$$

where $t_j$ is the feature obtained, and $*$ denotes the convolution operation, $b \in \mathbb{R}$ is the bias term, $\sigma$ is the activation function, such as $tanh$, and $e$ denotes the ordinal number, namely the $e$th word vector in the message $p_i$. Finally, by convolving all possible windows in the text vector $X$ using the convolution kernel $W$, the feature map of the text $p_i$ is obtained as $t = [t_1, t_2, \ldots, t_{L-k+1}]$ and $t \in \mathbb{R}^{L-k+1}$, where $t_1$ denotes the output features of the first sliding window in the CNN, $t_2$ denotes the output features of the second sliding window, and $t_{L-k+1}$ denotes the output features of the $L - k + 1$ th sliding window, after which the feature map $t$ is processed using a maximum pooling with step size $L - k + 1$, $\hat{t} = \max\{t\}$. In this paper, we apply sense field sizes of $k \in \{5, 6, 7\}$. After the maximum pooling operation, three feature vectors of length $d/3$ will be obtained, and then be spliced together to obtain the text $p_i$ and the final text content feature $m_i \in \mathbb{R}^d$, which will at last be spliced with the graph-structured feature of social organization networks.

**5. Exploratory Analysis of Organizational Geosocial Network with Graph Machine Learning**

*5.1. Experimental Deployment Environment*

In this paper, we completed organizational social network data integration, analysis, and machine learning model construction based on Python version 3.8; feature representation of text for network embedding with BERT; machine learning model (RF, KNN, LR) construction and model performance evaluation with Sklearn. We used DGL [44] for network dataset partitioning, graph construction, and graph neural network (GAT, GCN, MPNN) model construction, and PyTorch for deep learning model training and prediction.

The experiments were conducted on Google Colab platform with a Tesla P100 GPU. The pretrained BERT model has a dimension of 200 and was fine-tuned with a learning rateof $2 \times 10^{-5}$.

*5.2. Dataset Construction for Classification Task*

China's administrative divisions can be roughly divided into three levels: provincial, municipal, and county levels. With our database, we are able to pinpoint the social organizations and build a county-level OGN. As a country with a vast territory, China has thousands of county administrative divisions, forming a database of thousands of graph-structured data and ensuring us with sufficient data for training and testing machine learning models. The network of social organizations in southern Jiangsu in Figure 15 below shows us the tip of the iceberg of the database vividly.

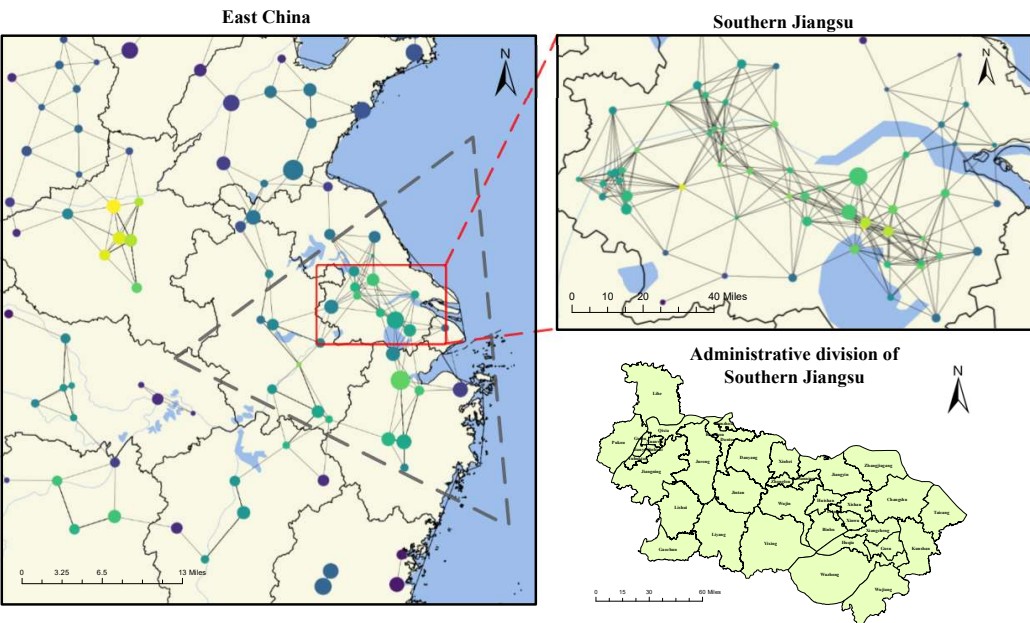

**Figure 15.** The organizational social network structure in the southern Jiangsu Province. The area framed by the triangle in the left side of the figure is the Yangtze River Delta.

In this paper, we selected three representative regions in China (Table 6): the Beijing–Tianjin–Hebei region, known as the "capital economic circle" of China, the Yangtze River Delta, which has experienced rapid economic development in recent years, and the Pearl River Delta region, which was the first to implement reform and opening up in China. With the three regions mentioned above as labels of the county-level OGN belong to them, machine learning models were trained for geographic-area-affiliation prediction task in these networks. Different regions have different development patterns under the influence of various factors such as economic, social, cultural, and geographical features, where the development of social organizations is embedded. If graph machine learning can effectively classify them, it can be a strong proof that graph machine learning models can map socioeconomic development patterns embedded in the network structure from an abstract dimension.

**Table 6.** Three regions used to construct the dataset.

| Region | Provincial Administrative Units | County Administrative Units |
|---|---|---|
| Beijing–Tianjin–Hebei Region | Beijing | – |
| | Tianjin | – |
| | Hebei | Baoding, Tangshan, Langfang, Shijiazhuang, Qinhuangdao, Zhangjiakou, Chengde, Cangzhou, Hengshui, Xingtai, Handan |
| Pearl River Delta | Guangdong | Guangzhou, Foshan, Zhaoqing, Shenzhen, Dongguan, Huizhou, Zhuhai, Zhongshan, Jiangmen |
| Yangtze River Delta | Shanghai | – |
| | Jiangsu Province | Nanjing, Wuxi, Changzhou, Suzhou, Nantong, Huaian, Yancheng, Yangzhou, Zhenjiang, Taizhou |
| | Zhejiang | Hangzhou, Ningbo, Wenzhou, Shaoxing, Jiaxing, Jinhua |

*5.3. Graph Attention Network Model Construction*

In this paper, we use graph attention network (GAT) to construct a neural network layer for representation learning of the embedding vector of the OGN structure, with the maximum aggregation-based READOUT function to aggregate the node features of the network, then input the results into the linear neural network layer and sigmoid activation function in turn to obtain the classification probability, so as to build a social organization–regional economic classification prediction model based on GAT.

As for the training and prediction process, we chose binary cross entropy as the loss function, Adam as optimizer, and the parameters are initialized with Xavier: the learning rate is $2 \times 10^{-5}$ , the dropout coefficient is set to 0.2, the batch size used for training is 16, the maximum number of iterations is 100, the number of layers of the graph attention network is 2, the dimension of the hidden layer is 256, and the coefficient of the $L_2$ regular term is $1 \times 10^{-3}$ during the training process.

$$h_i^{l+1} = \sigma \left( \sum_{j \in N_i} \alpha_{ij} h_i^l W^l \right) \tag{7}$$

where $h_i^l$ and $h_i^{l+1}$ are the vector representations of the $l$ and $l+1$ layer $i$ nodes, respectively; $N_i$ is the set of neighbor nodes of $i$ nodes; $a_{ij}$ is the number of attentional interrelationships between nodes $i$ and $j$; $W^l$ is the parameter matrix of the $l$th level; $\sigma$ is the nonlinear activation function.

The calculation procedure of $a_{ij}$ is shown in Equation (8).

$$\alpha_{ij} = \frac{\exp(e_{ij})}{\sum_{k \in N_i} \exp(e_{ik})} \tag{8}$$

where $e_{ij}$ is the edge vector representation of the connected nodes $i$ and $j$.

After the feature update of the nodes is completed by the GAT feature extraction layer, the node feature aggregation and model output are shown in Equations (9) and (10).

$$READOUT = \max \left\{ h_i^{l_n} \mid i \in N \right\} \tag{9}$$

$$y_{\text{output}} = \sigma(\text{Linear}(READOUT)) \tag{10}$$

*5.4. Evaluation Metrics*

In this paper, accuracy (Acc), F1-score, and precision are used as evaluation indicators, and the calculation of the indexes is shown in Equations (12) and (13).

$$\text{Recall} = \frac{TP}{TP + FN} \tag{11}$$

$$\text{Precision} = \frac{TP}{TP + FP} \tag{12}$$

$$\text{Accuracy} = \frac{TP + TN}{TP + TN + FP + FN} \tag{13}$$

$$F1 \text{ Score} = \frac{2 \times \text{Precision} \times \text{Recall}}{\text{Precision} + \text{Recall}} \tag{14}$$

$TP$ means the true positive case, indicating that the positive class is correctly predicted as the positive class; $TN$ means the true negative case, which means the negative class is correctly predicted as the number of negative classes; while $FP$ means the false positive case, indicating that the number of negative classes is incorrectly predicted to be positive; $FN$ means the false negative case, which means that the number of positive classes is incorrectly predicted to be positive.

### 5.5. Comparison Experiments with Baseline Models

In the geographic-area-affiliation prediction task, we constructed the GAT-based prediction model with three traditional machine learning models (RF, KNN, LR) and two graph neural network models (GCN, MPNN) as baseline models for comparison. The F1-score and accuracy results of the six models are shown in Figure 16 below.

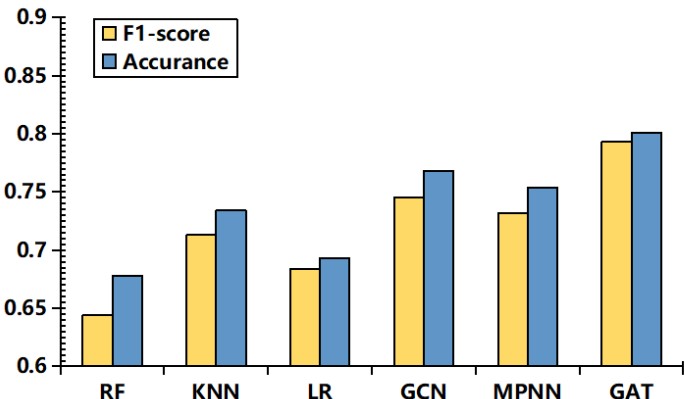

**Figure 16.** The experimental results.

### 5.5.1. Machine Learning Baseline Model

We chose random forest(RF), k-nearest neighbors(KNN) algorithm, and logistic regression(LR) as traditional machine learning baseline models. RF is an algorithm for building decision trees by using training data and random feature selection. RF performs multiple put-back sampling in the training set and builds a decision tree for each sampling result. KNN is a nearest neighbor algorithm for classification tasks [45] by finding $K$ nearest neighbor samples in the feature space of the samples to be classified and then deciding the class of the samples according to their class affiliation.

LR is a generalized linear regression analysis model [46] which constructs a linear hyperplane in the sample feature space by fitting the linear equation $y = \beta_0 + \beta_1 x_1 + \beta_2 x_2 + \cdots + \beta_n x_n$, dividing the feature space region into several sub-regions of categories so that each category of data belongs to the same sub-region, thus completing the classification task.

For the machine learning baseline models, the input network feature representations for model training are made by Node2vec [47].

The experimental results in Figure 16 show that the graph machine learning models has at least 8% performance improvement over the traditional machine learning model, mainly because traditional machine learning finds it difficult to learn complex semantic information, The RF model performs well in some simple classification tasks, but is prone to overfitting when it comes to complex data structures. The LR model alleviates the problem to some extent, but its performance improvement is not significant because it is limited by the linear classification space. The KNN model achieves relatively good results, which also reflects the importance of network structure from the side.

### 5.5.2. Graph Neural Network Baseline Model

We use graph convolution network (GCN) [29] and message passing neural network (MPNN) [48] to build a baseline model of a graph neural network for the organizational social network classification task. In the graph neural network baseline model, the structures of the aggregation and classification prediction models are consistent with the GAT-based prediction model except that GCN and MPNN are used for network structure feature extraction, respectively.

GCN is a classical graph neural network whose core idea is to transfer the image processing method based on convolutional neural network (CNN) to the graph structure

data and learn the relationship of the graph structure by aggregating the information around the nodes, and its update mechanism is shown in Equation (15).

$$h^{l+1} = \sigma\left(\tilde{D}^{-\frac{1}{2}}\tilde{A}\tilde{D}^{-\frac{1}{2}}h^l W^l\right) \tag{15}$$

where $\tilde{A}$ is $A + I$. $\tilde{D}$ is $D + I$, which represent the normalized adjacency matrix and degree matrix, respectively.

MPNN is a general computational framework of graph neural network that learns features from graphs through message passing, node updating, and aggregation, and can be independent of graph isomorphism. The update mechanism is shown in Equation (16).

$$h_i^{l+1} = U_l\left(h_i^l, \sum_{j \in N_i} M_l\left(h_i^l, h_j^l, e_{ij}\right)\right) \tag{16}$$

where, $U_l$ represents the update function; $M_l$ represents the message passing function.

The result shown in Figure 17 reveals that the accuracy of GAT is about 4% compared higher than other graph machine learning models on the OGN dataset. Classical graph machine learning is less effective than GAT due to the fact that GCN and MPNN are updated with full graph computation and the learned parameters are related to the complexity of the graph structure, while GAT uses attention coefficients point-by-point computation without relying on the Laplace matrix, which is more adaptive and has the ability to better utilize attention mechanisms to improve model performance based on syntactic dependencies, Compared with GCN and MPNN, the GAT-based model uses adaptive attention coefficients to represent the weights of edges between nodes, so that the neural network can pay attention to neighboring nodes with more influence (namely, larger weights) when nodes are updated, and learn more meaningful spatial and semantic information.

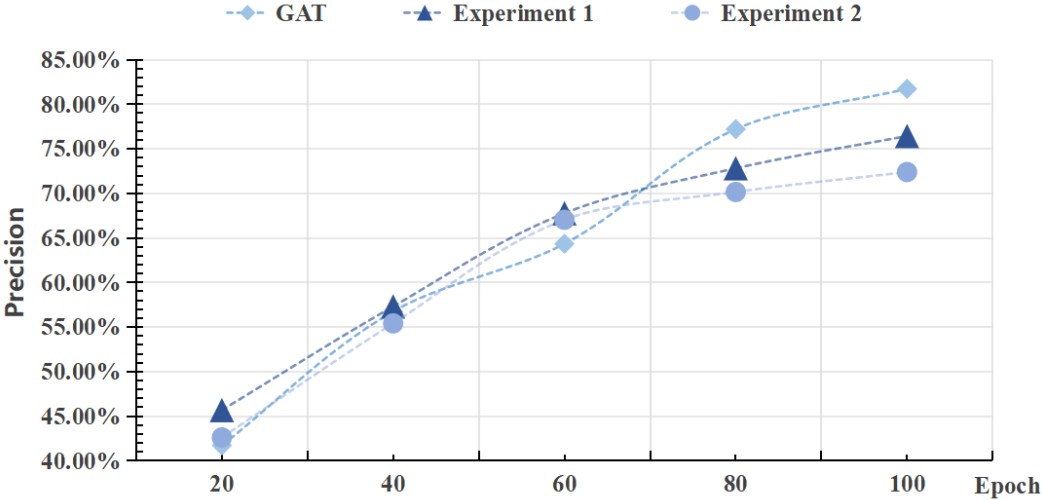

**Figure 17.** The experimental results.

It is clear that all six machine learning models have relatively good results for the prediction task, with the lowest one reaching an accuracy of 60%, which indicates that both deep-learning-based and traditional-machine-learning-based methods are able to learn the connection between organizational social networks and geographic, economic, and cultural factors. We hope that subsequent studies can be conducted with interpretable machine learning and thus go further in exploring the specific links between development patterns and geographic regions.

*5.6. Ablation Experiment*

In the field of artificial intelligence (AI), especially machine learning (ML), ablation refers to the removal of a component of an AI system [49]. Ablation study requires that the system exhibits graceful degradation: the system continues to function even if a component is lost or weakened. In the ablation experiment, we chose *precision* as the index to evaluate the performance of the model.

To further investigate the model's performance, two sets of ablation experiments were conducted on the proposed model on the OGN dataset: Experiment 1 used GloVe [50] with the same dimension of 200 in the word embedding layer instead; Experiment 2 used the multi-headed attention mechanism for model training in the encoding layer instead. The results of the ablation experiments are shown in Figure 17, from which it can be seen that in Experiment 1, the embedding layer used the same dimension of GloVe model for word embedding, and its accuracy differed significantly from that of the pretrained model BERT. Compared with GloVe, the fine-tuned BERT is more effective in capturing the semantic information of the text, i.e., accurate semantic information extraction plays an important role in improving the performance of the model. In Experiment 2, with the adoption of the multi-head, the reason that the effect was not improved after the attention mechanism is that when the OGN structure contains multiple aspect targets, the attention mechanism may focus the socioeconomic embedding on the wrong aspect target, further illustrating the importance of the information of the network structure as a whole in the classification task.

## 6. Conclusions

Society is a complex system whose development comes from the collision and convergence of different social entities. In this paper, we construct a novel database of social organizations in China with related information, using the open data platform provided by the Ministry of Civil Affairs of the People's Republic of China, which, to our knowledge, is one of the few social organizational databases that have been applied to computational social science research. We believe that the construction of this database can provide more and more powerful help for researchers to explore the development of Chinese social organizations and the macro changes of Chinese society in the future.

With the database, we explored the network structure composed of social organizations and related social entities. We proposed four types of social organization networks based on graph theory, trying to structuralize the development patterns of social organizations in different regions, which are characterized by local policy, economic, and cultural factors. We construct a graph-model-based organizational geosocial network(OGN), with the help of natural language processing(NLP) technology to embed the textual information into the network, which enables it to fuse more dimensions of information, thus representing richer structural and semantic features of the complex network.

Using machine learning models, we conducted exploratory research on the relationship between the development patterns of organizational social networks and the geographic zones to which they belong. Our machine learning models achieved relatively good results on the training data, with an average accuracy rate of 70%. However, it is important to emphasize that our aim is not simply to pursue the accuracy or to create a new state of the art (SOTA), but to explore the correlation between the graph-structured network data and the socioeconomic differences embedded in geographic space through the geographic-area-affiliation prediction task.

In future research, we hope to build larger and more complex graph network structures from a multidimensional perspective [51,52], and we also hope to highlight the role of interpretable machine learning [53] to decrease the black box nature of deep learning and help us gain an in-depth understanding of the causal relationship between the development of social organization and relevant policy, economic, and cultural factors.

**Author Contributions:** Conceptualization, Xinjie Zhao, Hao Wang, and Shiyun Wang; methodology, Xinjie Zhao and Hao Wang; validation, Xinjie Zhao, Shiyun Wang, and Hao Wang; formal analysis, Shiyun Wang and Hao Wang; data curation, Xinjie Zhao; writing—original draft preparation, Xinjie Zhao and Shiyun Wang; writing—review and editing, Xinjie Zhao, Hao Wang, and Shiyun Wang; visualization, Xinjie Zhao and Shiyun Wang. All authors have read and agreed to the published version of the manuscript.

**Funding:** This research is supported by the Youth Project of the National Social Science Foundation of China "Research on Unbalanced and Insufficient Development of Social Organizations Based on Big Data Method" (20CSH089).

**Institutional Review Board Statement:** Not applicable.

**Informed Consent Statement:** Not applicable.

**Data Availability Statement:** Since our database has just been developed and the amount of data is huge, there are still some instability factors in it. Therefore, after some more in-depth testing, we will look for a suitable opportunity to publish it, and if you are interested in the database, you can also contact us directly to collaborate on the research.

**Conflicts of Interest:** The authors declare no conflict of interest.

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
