# Peer review of "Organizational Geosocial Network: A Graph Machine Learning Approach Integrating Geographic and Public Policy Information for Studying the Development of Social Organizations in China"

_ijgi, doi:10.3390/ijgi11050318_

Round 1

Reviewer 1 Report

The authors provided the database of social organizations in China, modeled the development of social organizations, and provided a graph attention network. Results are very promising, and so is the topic, which is worth researching and relevant to the field of the ISPRS International Journal of Geo-Information.

There are some things that must be addressed before publication.

Did the authors draw Figure 2? If so, which data was used? Figure 2 must be better explained - what do different colors mean?

All figures must be mentioned in the text. Almost none of them is mentioned in this version of the paper.

In which format is the newly created database? Is the database somewhere available/published? 

How long did the scraping process last? How big is the newly created database? How many entries does it have? The process of retrieving data must be much better presented. 

How did the authors decide on the parameters settings, e.g., batch size, number of iterations, etc.? This must be much better described.

Even though the authors provided some basic evaluation metrics, i.e., F1-score and accuracy, I suggest that they also conduct a statistical analysis.

More in-depth analysis of the results is missing.

Otherwise, the paper is well-written. Used references are relevant and up-to-date.

Author Response

The response letter to the reviewers

We authors of this manuscript would like to thank the anonymous reviewers’comments and the editor’s suggestion on the earlier manuscript of this paper. We have carefully revised this manuscript according to all comments as follows (sentences in bode are our replies).

“ Did the authors draw Figure 2?

If so, which data was used?

Figure 2 must be better explained - what do different colors mean?”

 A:

Our intention was to present the organizational-geographic network of China in a more visual form, so we did not modify Figure 2 much in the first version of this paper. However, in the revised version, we have actively listened to your comments, and in order to enhance the rigor and readability of the academic paper, we have revised the original figure in more detail to make it richer, clearer, and to some extent more explanatory.

“ In which format is the newly created database?

 Is the database somewhere available/published? 

 How long did the scraping process last?

 How big is the newly created database?

 How many entries does it have?

 The process of retrieving data must be much better presented. “

  A:

We would like to answer questions relevant to the database uniformly. Based on the reviewers' queries and suggestions, we, the authors, pondered the third part of the paper related to database construction and descriptive statistical analysis. There were some rather vague areas in the article which have been restructured and added more details in the revised manuscript after absorbing opinions. We sincerely hope this revision will enable readers to better understand the practical ideas in this paper.

In particular, the third part of the article has been systematically modified and divided into three subsections. The first part is the Design and Implementation of Web Crawlers. A crawler program was designed through Python to crawl web data and store the acquired data in CSV format in a cloud server to facilitate our calls when analyzing the data.

Owing to the fact that a million-dollar database is relatively massive and greatly affected by network environment as well as equipment, we adopted a multi-threaded approach and crawled for a month using three computers to eventually integrate into a more complete database. This database we built is unique with a total of 1.09 million social organizations.

In this section, the process of collecting and analyzing specific data information is vividly demonstrated, which it is considered interesting and enlightening, however, for some legal reasons, we do not publish the specific encryption of the website. The encryption and decryption details are not published during our data mining process.

The remaining two sections provide descriptive statistics of the data from the perspectives of natural language processing and geographic information, respectively, which are the two aspects we want to integrate and explore in this paper. Descriptive statistics make further exposition of the trends and specific characteristics about social organizations in China.

Speaking of data accessibility, since our database has just been developed and the amount of data is enormous, there are still some instability factors. Therefore further data cleaning is still underway. After the processing is completed and some more intensive testing is done, we will find a suitable opportunity to publish it, or if you are more interested in the database, you can contact us directly and we can collaborate on the research together.

“ How did the authors decide on the parameters settings, e.g., batch size, number of iterations, etc.? This must be much better described.

Even though the authors provided some basic evaluation metrics, i.e.,

F1-score and accuracy,

I suggest that they also conduct a statistical analysis.

More in-depth analysis of the results is missing. ”

 A:

In the experimental design and implementation section of the first edition, we have streamlined the content to a certain extent for reasons of space and readability, but this idea may cause some inconvenience to the reader's understanding.

As authors, we acknowledge that the experimental design was a challenge for us, as, on one hand, we tried to demonstrate the potential of applications of graphical neural networks in OGN, but on the other hand, we wanted the experimental and algorithmic design to be as brief and easy to understand as possible, so as not to make this section an intimidating algorithmic article. Therefore we adopted classification, a classical research tool in machine learning, as the experimental task.

In this revised version, the specific details of the experiments and the analysis of the experimental results are explored in depth, and the depiction of each parameter of the model is depicted in relative detail. For the baseline experiments, we distinguish traditional machine learning algorithms from graph depth learning algorithms with clearer descriptions, and the characteristics of different algorithms and experimental results are elaborated in more detail. Based on this, this revision adds an ablation experiment, which we hope will help us better understand the detailed mechanisms of graph attention networks, the classification of social organizations, and the extraction of abstract features of their social and geographical embeddings. The abstract features of the geographic embedding are extracted.

Reviewer 2 Report

The authors develop an analysis framework for studying the development of social organizations in China. The matter is interesting and the developed Chinese social organizations database is a foundational work for other analytical work. The integration of geographic information and public policy. The paper suffers the following small limits, authors should carefully revise this manuscript before its publication: 

  1. In Sec. 3, Figure 10 does not appear in the article. Therefore, what is the point of mentioning the NPOs in China?
  2. In Sec. 3, not only contains the proposed analysis and processing work, but also has some background knowledge and related work (e.g., the concept of NLP from line 285 to 300, the concept of machine learning from line 309, and so on). I wonder why not put these in Sec. 2.
  3. In Sec. 4, the title of Table 4 is incorrect.
  4. In Sec. 5, the experimental part is a little bit weak. As I see, only Sec. 5.4.4 is about the performance evaluation and from Sec. 5.1 to Sec. 5.4.3 are experiment settings. If the authors could add more experimental analysis, that would be better. Moreover, according to Figure 6, why does the GAT-based prediction model outperform others?

Author Response

The response letter to the reviewers

We authors of this manuscript would like to thank the anonymous reviewers’comments and the editor’s suggestion on the earlier manuscript of this paper. Wehave carefully revised this manuscript according to all comments as follows (sentences in bode are our replies).

“ In Sec. 3, Figure 10 does not appear in the article. ”

 A:

 The structure of the article was adjusted more precisely and standardized in the revision. The figure mentioned serves in the paper to better show the reader the geospatial properties of the distribution of social organizations, however, without being interpreted well in the structuring process of the paper, due to negligence in the first edition. As suggested by the reviewer, we have made structural improvements in the construction and the fundamental descriptive statistics of the database. In this edition, each figure can be corresponding to the text, aiming to bring a more exquisite reading experience to readers.

“Therefore, what is the point of mentioning the NPOs in China?”

 A:

Thank you for pointing this out. It is relatively unclear how the Chinese government defines social organizations with respect to the process of social organization development leading to the appearance of the NPOs and social organization, two words that may cause ambiguity in the formulation of the paper. In the revised version we have unified expressions that may be ambiguous and the presentation of social organization throughout the text in order to pursue unity and wholeness of the whole article.

“In Sec. 3, not only contains the proposed analysis and processing work, but also has some background knowledge and related work (e.g., the concept of NLP from line 285 to 300, the concept of machine learning from line 309, and so on). I wonder why not put these in Sec. 2.”

A:

For the construction of the database and descriptive statistics related to the data, we actively listened to the opinions from various parties. The third part of the article was systematically revised and divided into three modules, the first of which is the design of the crawler program and the data mining strategy.

There were some rather vague areas in the article which have been restructured and added more details in the revised manuscript after absorbing opinions. We sincerely hope this revision will enable readers to better understand the practical ideas in this paper.

The section on the basics of natural language processing has been integrated into the Related Topics in Sec. 2., making it possible to have a clearer and more standardized presentation of the entire database construction and the underlying statistical descriptions in section 3.

“ In Sec. 4, the title of Table 4 is incorrect.”

A:

We feel sorry for our carelessness. In our revised manuscript, the error have been corrected.

“ In Sec. 5, the experimental part is a little bit weak. As I see, only Sec. 5.4.4 is about the performance evaluation and from Sec. 5.1 to Sec. 5.4.3 are experiment settings. If the authors could add more experimental analysis, that would be better.  Moreover, according to Figure 6, why does the GAT-based prediction model outperform others?”

A:

In the experimental design and implementation section of the first edition, we have streamlined the content to a certain extent for reasons of space and readability, but this idea may cause some inconvenience to the reader's understanding.

As authors, we acknowledge that the experimental design was a challenge for us, as, on one hand, we tried to demonstrate the potential of applications of graphical neural networks in OGN, but on the other hand, we wanted the experimental and algorithmic design to be as brief and easy to understand as possible, so as not to make this section an intimidating algorithmic article. Therefore we adopted classification, a classical research tool in machine learning, as the experimental task.

In this revised version, the specific details of the experiments and the analysis of the experimental results are explored in depth, and the depiction of each parameter of the model is depicted in relative detail. For the baseline experiments, we distinguish traditional machine learning algorithms from graph depth learning algorithms with clearer descriptions, and the characteristics of different algorithms and experimental results are elaborated in more detail. Based on this, this revision adds an ablation experiment, which we hope will help us better understand the detailed mechanisms of graph attention networks, the classification of social organizations, and the extraction of abstract features of their social and geographical embeddings. The abstract features of the geographic embedding are extracted.

Reviewer 3 Report

  • There is no description in the text, for Figure 1 - 8, 10, 12-15. It is ambiguous where the figure in the paper corresponds to the text.
  •  In the case of a map such as Figure 2 and 10, basic elements such as direction and scale must be added.
  •  It seems inappropriate for the figure 10,11, table 2 to contain Chinese characters.
  • A more detailed explanation of how much data was collected and analyzed is needed.

Author Response

The response letter to the reviewer

We authors of this manuscript would like to thank the anonymous reviewers’ comments and the editor’s suggestion on the earlier manuscript of this paper. We  have carefully revised this manuscript according to all comments as follows (sentences in bode are our replies).

"There is no description in the text, for Figure 1 - 8, 10, 12-15. It is ambiguous where the figure in the paper corresponds to the text."

A: 

First of all, we, the authors, apologize for some oversights and mistakes in the illustrations of this paper and the bad reading experience they caused, which we must admit is one of our failures in writing the paper. We have corrected the problems of the illustrations in the paper in this edition.

We have checked all the illustrations in this article to make sure that they are not connected to the content of the article, and we have revised the article to make sure that each illustration is reflected in the article. Each illustration is reflected in the text in the revised version so that the context is natural and coherent enabling the reader to understand the idea of the text more clearly.

 "In the case of a map such as Figure 2 and 10, basic elements such as direction and scale must be added."

A: 

The suggestions on the specification of geographic illustrations have contributed greatly to improving the standardization and rigor of our paper. We have consulted the appropriate professionals and have proactively corrected our errors, and we believe that the illustrations are now able to meet the basic geographic norms.

" It seems inappropriate for the figure 10,11, table 2 to contain Chinese characters."

A: 

Thank you very much for your advice on this detail.

For us authors, since Chinese is our native language, we may have missed the translation of some critical information in Chinese, which has caused some unnecessary obstacles to the reading process.

We have thoroughly checked the Chinese details that appear in the illustrations and tables of the paper this time.

We believe that it is necessary to keep the original Chinese text in order to give the reader a clearer understanding of some specific and subtle information about Chinese social organizations.  At the same time, we have thoroughly translated these Chinese contents and attached the translated contents around the corresponding Chinese contents to ensure their readability, which we hope will help readers to cross the barrier between languages.

"A more detailed explanation of how much data was collected and analyzed is needed."

 A: 

For the construction of the database and descriptive statistics related to the data, we actively listened to the opinions from various parties. The third part of the article was systematically revised and divided into three modules, the first of which is the design of the crawler program and the data mining strategy.

 In this section, we vividly demonstrate the process of collecting and analyzing specific data information, which we found interesting and inspiring, but for some legal reasons, we did not publish the exact encryption of the website. We do not publish the details of encryption and decryption in the data mining process.

The other two sections provide descriptive statistics of the data from the perspectives of natural language processing and geographic information, respectively, which are the two aspects we would like to integrate and explore in this paper. We believe that the descriptive statistics from these two perspectives can make the further exposition of the trends and specific characteristics of social organizations in China.
